# Carbon Monoliths with Hierarchical Porous Structure for All-Vanadium Redox Flow Batteries

**Jose Francisco Vivo-Vilches** [1,*,†], **Blagoj Karakashov** [2], **Alain Celzard** [2], **Vanessa Fierro** [2], **Ranine El Hage** [1], **Nicolas Brosse** [3], **Anthony Dufour** [4] **and Mathieu Etienne** [1,*]

[1] Laboratoire de Chimie Physique et Microbiologie pour les Matériaux et l'Environnement (LCPME), Université de Lorraine, Centre National de la Recherche Scientifique (CNRS), F-54000 Nancy, France; ranine.el-hage@univ-lorraine.fr

[2] Institute Jean Lamour (IJL), Université de Lorraine, Centre National de la Recherche Scientifique (CNRS), F-88000 Épinal, France; bkarakashov@gmail.com (B.K.); alain.celzard@univ-lorraine.fr (A.C.); vanessa.fierro@univ-lorraine.fr (V.F.)

[3] Laboratoire d'Etudes et de Recherche sur le Matériau Bois (LERMAB), Université de Lorraine, F-54000 Nancy, France; nicolas.brosse@univ-lorraine.fr

[4] Laboratoire Réactions et Génie des Procédés (LRGP), Université de Lorraine, Centre National de la Recherche Scientifique (CNRS), F-54000 Nancy, France; anthony.dufour@univ-lorraine.fr

[*] Correspondence: jvivo@ing.uc3m.es (J.F.V.-V.); mathieu.etienne@univ-lorraine.fr (M.E.); Tel.: +34-91-6-24-99-90 (J.F.V.-V.); Fax: +33-3-83-275-444 (J.F.V.-V.)

[†] Present address: Materials Science and Engineering and Chemical Engineering Department, School of Engineering, Universidad Carlos III Madrid, ES-28911 Madrid, Spain.

**Abstract:** Carbon monoliths were tested as electrodes for vanadium redox batteries. The materials were synthesised by a hard-templating route, employing sucrose as carbon precursor and sodium chloride crystals as the hard template. For the preparation process, both sucrose and sodium chloride were ball-milled together and molten into a paste which was hot-pressed to achieve polycondensation of sucrose into a hard monolith. The resultant material was pyrolysed in nitrogen at 750 °C, and then washed to remove the salt by dissolving it in water. Once the porosity was opened, a second pyrolysis step at 900 °C was performed for the complete conversion of the materials into carbon. The products were next characterised in terms of textural properties and composition. Changes in porosity, obtained by varying the proportions of sucrose to sodium chloride in the initial mixture, were correlated with the electrochemical performances of the samples, and a good agreement between capacitive response and microporosity was indeed observed highlighted by an increase in the cyclic voltammetry curve area when the $S_{BET}$ increased. In contrast, the reversibility of vanadium redox reactions measured as a function of the difference between reduction and oxidation potentials was correlated with the accessibility of the active vanadium species to the carbon surface, i.e., was correlated with the macroporosity. The latter was a critical parameter for understanding the differences of energy and voltage efficiencies among the materials, those with larger macropore volumes having the higher efficiencies.

**Keywords:** vanadium redox flow battery; hierarchical carbon; carbon electrode; porosity; sucrose-based carbon monolith

## 1. Introduction

Due to the problem of worldwide carbon dioxide emissions and climate change, an increasing effort to replace fossil fuel-derived energy sources by renewable ones, such as solar or wind, has been made in the last years [1–5]. Nevertheless, the main problem of this type of energy generation system is their dependence on climatic circumstances and, therefore, devices able to store the energy when it is generated and to release it on demand have been developed [1–3,5–9]. All of them present differences in terms of amount of energy that can be stored, power that can be supplied, long-term stability and cyclability.

For stationary applications, a large amount of energy needs to be stored, and other energy storage devices such as supercapacitors [3,10–12] or Li-ion batteries [13–16] are not suitable (in the case of the latter, large Li-ion batteries will be very inefficient and, since organic solvent are used, power surge could lead to explosions). In this sense, redox flow batteries (RFB) are presented nowadays as sound alternatives [17–21].

Redox flow batteries are composed of a stack of several cells connected in series, each of them containing two electrodes (cathode and anode) separated by an ion-exchange membrane [17–24]. Electrolytes (catholyte and anolyte) are contained in separated tanks and pumped through the system to be recirculated. In this case, the energy is not stored in the electrodes when the battery is charged, but in the electrolytes, as a consequence of the change of oxidation states of the active species. This fact improves the flexibility of the design and makes it easier to stop the system in case of failure without losing the stored energy [25]. Furthermore, RFBs present better performances in terms of cost, long-life cycle and energy efficiency [17–19,26]. It can be said that an RFB is a special kind of fuel cell wherein the species responsible for the energy production are not consumed but regenerated and stored during the charge and discharge of the device.

Several electrolytes have been tested and, for instance, flow batteries based on iron/chromium [27], polysulfide/iodide [28], quinone/bromine [29] or zinc/cerium [30] have been developed. The main problem of RFBs based on this kind of mixed electrolyte is the possibility of irreversible cross-contamination between catholyte and anolyte after long-term cycling. To overcome this drawback, electrolytes based on dissolved vanadium salts were developed and are still the most common ones because, thanks to the different oxidation states of vanadium (II, III, IV and V), both compartments can be filled with the same electrolyte, a stoichiometric mixture of V(III) and V(IV) [6–8,26,31–35]. Several works can be found in the literature about vanadium redox flow batteries (VRFB) in which the composition of the electrolyte and the use of additives were studied to improve the performances and the lifetime of the batteries [36,37] in addition to the optimization of the different components of the VRFB such as the membrane and the electrode materials, especially those based on carbon or graphite felts, in order to increase the efficiency of the system.

In fact, different carbon materials such as carbon felts or carbon paper have been extensively explored and compared as electrodes for VRFB [6–8,38–41]. Their main drawback is the relatively slow charge transfer kinetics that they present, so several approaches have been proposed for improving their performance, such as: Thermal or chemical treatments to increase their hydrophilicity [42,43]; combining these materials with others such as carbon nano-forms, activated carbons or metal oxide nanoparticles was reported to overcome this problem [8,44–48]. A good adhesion of the active material to the carbon felt or carbon paper support is necessary to avoid its removal and to guarantee its use over prolonged periods. Activation of carbon felt or covalent functionalisation with various heteroatoms such as oxygen or nitrogen also proved to be effective for improving the VRFB performances [49–51].

Nevertheless, other carbon materials and especially hierarchical porous ones have not been studied enough, even though they have proven to be relevant in other electrochemical applications [10,52–54] and even though several synthetic routes for obtaining them, from bioresources such as sugars or tannins, have been developed [55–57]. Recently, there has been an increased interest in the field of hierarchical porous carbons [58,59] since traditional activated carbons are known as materials with highly developed porosity but insufficient control of pore accessibility. In the field of RFBs, the kind of electrodes commercially used are ones with very regular porosity (carbon felts), precisely to avoid tortuosity and, therefore, to improve the flowing of the solution and the accessibility of active species to the entire surface of the electrodes. Hard-template and soft-template routes are very useful methods to obtain carbons with hierarchical porosity that can be applied to a broad range of precursors for obtaining specific materials in terms of porosity and surface chemistry [13,53,57,60].

In the present work, monolithic carbon materials based on sucrose polycondensed in the presence of sodium chloride (NaCl) were prepared with a protocol similar to the one described elsewhere [57]. Once obtained, the materials were characterised in-depth in terms of porosity (gas adsorption, mercury intrusion porosimetry and scanning electron microscopy), carbon nanotexture (Raman spectroscopy) and composition (elemental analysis). After that, their relevance to promote vanadium redox reactions was evaluated by cyclic voltammetry. Finally, they were tested as electrodes for VRFBs, showing promising performances for being used as electrodes for this application.

## 2. Materials and Methods

### 2.1. Reagents

Sucrose ($\geq$99.5%) was supplied by Sigma-Aldrich (St. Louis, MO, USA); sodium chloride (NaCl, $\geq$99.9%) by VWR Chemicals (Radnor, PA, USA); and vanadium trichloride (VCl$_3$, 97%), vanadium oxysulphate hydrate (VOSO$_4$·xH$_2$O, x = 3–5, 97%) and sulphuric acid (H$_2$SO$_4$, 95–98%) by Sigma-Aldrich.

### 2.2. Carbon Monoliths Preparation

Sucrose-based monoliths were synthesised by a protocol similar to the one reported by Wilson et al. [57] with some modifications. A scheme of the synthesis steps is depicted in Figure 1. In a first step, sucrose and NaCl were mixed in the appropriate sucrose to NaCl weight ratio (0.80, 1.00 or 1.25) and ball-milled for 2 h at 200 rpm in the presence of acetone, always employing the same solid to acetone (3:1) and reagents to agate balls (2:1) weight ratios. Once milled, the mixture was dried in a furnace at 60 °C, and next the temperature was increased to 185 °C to produce the melting of sucrose (Step 2). The mixture was removed from the furnace from time to time for homogenisation and, once softened, left in the furnace for five more minutes to ensure the completion of the melting process. Then, it was transferred to a stainless steel mould with inner dimensions 10.0 × 10.0 × 2.5 cm$^3$, sealed beforehand with vulcanised silicone resistant to high temperature and pressure. The mould filled with the molten mixture was then covered with a stainless–steel square lid and put into a hot press, where the compression was performed at 185 °C and 18 bar for 3 h (Step 3). After that, the press was opened and the mould with the sample inside was left to cool overnight at room temperature. The sample was then removed from the mould and introduced into a furnace under air at 200 °C for 2 h for annealing, so the structure was stabilized prior to being pyrolyzed (Step 4). Then, a first pyrolysis was performed under inert atmosphere (N$_2$) at 750 °C (heating rate 1 °C min$^{-1}$) for 2 h (Step 5) before removing the hard template (NaCl) to avoid the lixiviation of the material during the next step (since at this point, caramelized sucrose is still soluble in water). For creating the macroporosity, the pyrolyzed sample was immersed in hot distilled water for several days (Step 6), thus leaching NaCl out, until no more chloride was detected in the rinsing water (checked by AgCl precipitation). The last step consisted in the pyrolysis at higher temperature (900 °C, 1 °C min$^{-1}$, 2 h) to get the final materials which were labelled SucNaClX, where X indicates the sucrose to NaCl weight ratio (0.80, 1.00 or 1.25).

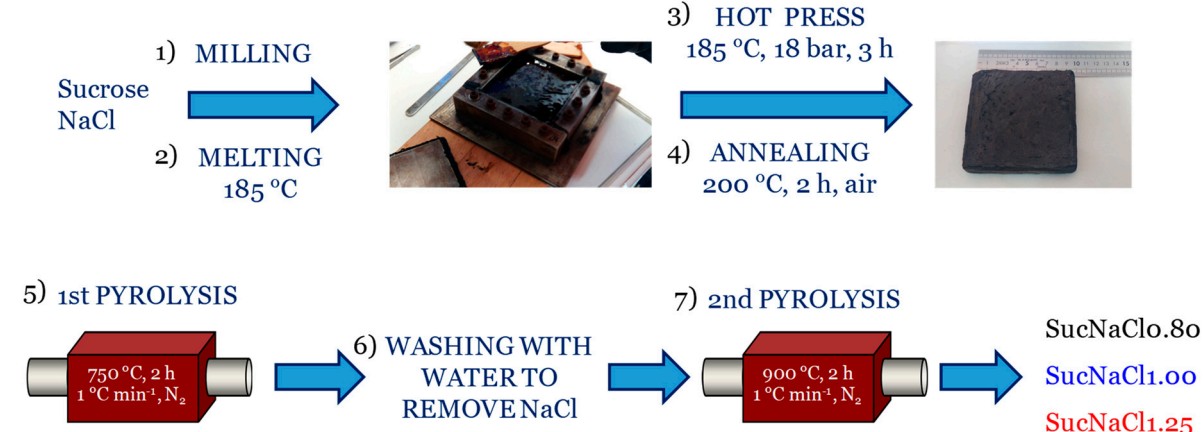

**Figure 1.** Scheme of the synthetic pathway for preparing sucrose-based monoliths, SucNaClX, where X stands for the sucrose to NaCl weight ratio.

### 2.3. Porosity Assessment and Mechanical Characterisation

For the obtention of nitrogen adsorption isotherms at $-196\,^{\circ}$C and the carbon dioxide one at $0\,^{\circ}$C, automatic analysers ASAP 2020 and ASAP 2420 (Micromeritics, Norcross, GA, USA) were used, respectively. Those isotherms were then analysed to obtain the BET surface area ($S_{BET}$), the total adsorbed nitrogen according to Gurvitch's rule ($V_{0.97}$), the micropore volume ($W_0$) by the Dubinin–Radushkevich method, the average micropore width ($L_0$) from Stoeckli's equation and the pore size distribution (PSD) in the micro-mesopore range as well as the corresponding surface area ($S_{NLDFT}$) by applying the NLDFT theory. Mercury intrusion porosimetry was also carried out with a Micromeritics AutoPore IV 9500 device to obtain the bulk density ($\rho_{Hg}$) of the materials, as well as the pore size distribution in the meso-macropore range. Scanning Electron Microscopy (SEM) micrographs were obtained at the PLateforme Aquitaine de Caractérisation des MATériaux, PLACAMAT (Bordeaux, France) using an EVO 50 instrument, equipped with an Everhart–Thornley secondary electron detector and operating at a maximum voltage of 30 kV. Mechanical properties were finally evaluated by compression using an Instron 5944 universal testing machine equipped with a 2 kN head and applying a load rate of 2 mm·min$^{-1}$. The corresponding stress–strain curves of the materials allowed calculating the compressive strength ($\sigma_c$) and the Young's Modulus $E$.

### 2.4. Chemical Characterization

The surface chemistry of the materials was investigated using Raman spectroscopy, using a confocal Raman microscope (inVia® Qontor with a Peltier cooled CCD camera, Renishaw, Wotton-under-Edge, UK). The laser in this instrument irradiates at a wavelength of 532 nm and with an irradiance $< 5$ kW cm$^{-2}$, while it contains an objective X50 (numerical aperture, 0.55; grating, 1200 lines mm$^{-1}$; spectral resolution about 3 cm$^{-1}$ when obtaining 10 acquisitions of 4 s). Since heterogeneities can be present at the surface, spectra were obtained at ten different points of the materials' surface, and the spectra plotted for each sample was the average of these ten. Then intensities for the G and the D band ($I_G$ and $I_D$), as well as their ratios, were obtained, as is commonly done for carbon materials. The overall composition was determined by elemental analysis, using a Vario EL Cube (Elementar, Langenselbold, Germany) elemental analyser according to two separate steps: One for dosing C, H, N and S together, and another one for the determination of the oxygen content in the samples.

### 2.5. Cyclic Voltammetry and Charge–Discharge Tests

To evaluate the performance of the materials for promoting vanadium electrochemistry, cyclic voltammetry (CV) curves were obtained with a potentiostat (SP 50, Biologic

Instruments, Seyssinet-Pariset, France). A three-electrode setup was used: As reference electrode (RE), Ag/AgCl 3 M KCl; as counter electrode (CE), graphite rod; and as working electrode (WE), a piece of monolithic sample previously weighed was pinned by with a graphite tip that was stuck to a copper rod using copper adhesive tape. To register the CV curves, the potential was swept from −1.0 to +1.5 V at a scan rate of 5 mV·s$^{-1}$. Preparation of the electrolyte was as follows: A stock solution of 1.5 M V(III)/V(IV) in 3 M $H_2SO_4$ was prepared and then diluted ten times with 3 M $H_2SO_4$ to get the final concentration (0.15 M). The stock solution (1.5 M) was also employed as the battery electrolyte in charge–discharge experiments for both the positive and the negative half cells.

For the battery system, a homemade cell was employed which was connected to a potentiostat (SP 150 equipped with a 20 A Booster, Biologic Instruments). The cell comprised two symmetrical and identical compartments separated by a Nafion® 115 membrane (Ion Power, München, Germany), each one containing a graphite composite collector. Each compartment had an empty volume of $2.0 \times 7.0 \times 0.4$ cm$^3$ where the electrodes were placed, so two pieces were cut from each monolith and fitted into them. Two separated tanks, each of them containing 20 mL of the electrolyte, were used to feed both half cells of the system. A peristaltic pump was used to force the electrolyte to pass through the electrodes before returning to the tank at a flow rate of 20 mL·min$^{-1}$. As a first test, polarization curves were obtained by changing the voltage from 0.0 to 1.1 V and registering the current. After that, galvanostatic charge–discharge experiments (GCD) were performed at several current densities (20, 50 and 80 mA·cm$^{-2}$) and potential limits were increased accordingly (1.05 to 1.65 V, 0.80 to 1.90 V and 0.60 to 2.10 V).

## 3. Results and Discussion

### 3.1. Porosity of the Carbon Monoliths

Nitrogen and carbon dioxide adsorption isotherms at −196 °C and 0 °C, respectively, are shown in Figure 2, and the most interesting characteristics deduced from them are summarised in Table 1. All samples presented type Ia adsorption isotherms, with a large uptake at very low relative pressures corresponding to micropore filling, and almost no increase at relative pressure ($P/P_0$) above 0.1, suggesting that these materials are essentially microporous. Both the micropore volume obtained by applying the Dubinin–Radushkevich model to nitrogen adsorption isotherm ($W_0$ ($N_2$)) and the BET area ($A_{BET}$) increased with the proportion of sucrose in the initial formulation, evidencing the corresponding increase of microporosity.

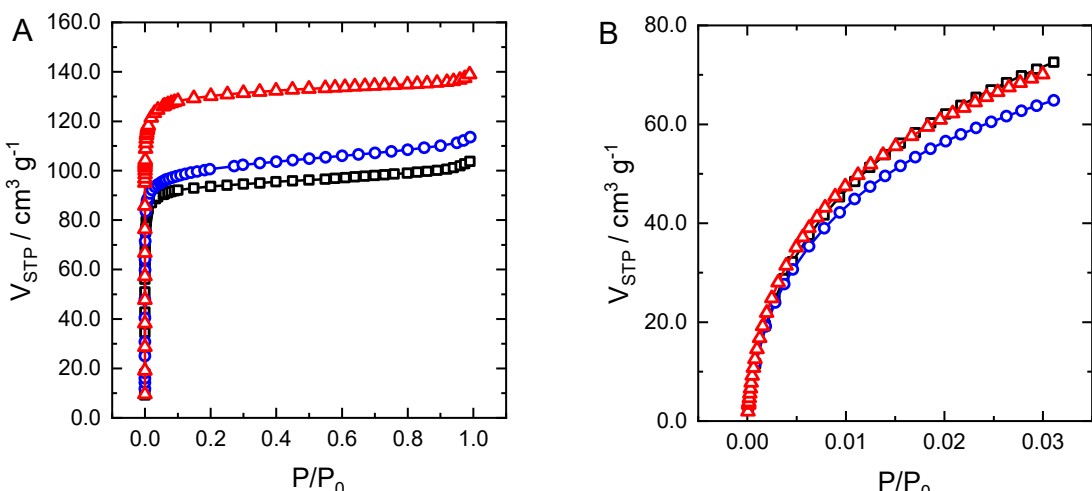

**Figure 2.** (**A**) Nitrogen adsorption isotherms at −196 °C, and (**B**) carbon dioxide adsorption isotherms at 0 °C for samples SucNaCl0.80 (□), SucNaCl1.00 (○) and SucNaCl1.25 (△).

**Table 1.** Porosity characteristics of the materials deduced from gas adsorption experiments, and bulk density from mercury intrusion porosimetry.

| Sample | $S_{BET}$ $m^2 g^{-1}$ | $W_0$ ($N_2$) $cm^3 g^{-1}$ | $W_0$ ($CO_2$) $cm^3 g^{-1}$ | $L_0$ ($N_2$) nm | $S_{NLDFT}$ $m^2 g^{-1}$ | $V_{0.97}$ $cm^3 g^{-1}$ | $\rho_{Hg}$ $g cm^{-3}$ |
|---|---|---|---|---|---|---|---|
| SucNaCl0.80 | 377 | 0.155 | 0.236 | 1.01 | 744 | 0.159 | 0.16 |
| SucNaCl1.00 | 400 | 0.177 | 0.224 | 1.29 | 867 | 0.173 | 0.19 |
| SucNaCl1.25 | 527 | 0.201 | 0.273 | 0.62 | 797 | 0.212 | 0.21 |

It is not surprising that a larger amount of carbon precursor (sucrose) in the synthesis led to a more microporous carbon structure after pyrolysis, since micropores are formed during this step. Regarding the average micropore width ($L_0$ ($N_2$) in Table 1) and the pore size distribution obtained by applying NLDFT to the adsorption isotherms (Figure 3A), it can be observed that the sample with the highest fraction of "broad" micropores corresponds to the one with a 1:1 sucrose to NaCl weight ratio (SucNaCl1.00). A lower sucrose proportion in the precursor mixture led to a material with a lower amount of "wide" micropores (width around 1 nm according to PSD, Figure 3A), while for the material with the highest sucrose content, the peak centred at around 1 nm disappeared from the PSD curve, and only peaks centred on 0.36 and 0.60 nm remained. Consequently, a marked reduction of average micropore width ($L_0$ ($N_2$)) is observed for this sample. Considering that carbon dioxide is preferentially absorbed in the narrowest micropores at 0 °C, it is thus logical to find that $W_0$ ($CO_2$) is the minimum for the sample with wider micropores (SucNaCl1.00) and is the highest for the sample SucNaCl1.25, which is the one with the lowest average micropore width and the highest micropore volume. These increases in microporosity will be relevant to explain the changes observed in CV curves for the capacitive signals of our samples. Nevertheless the relevant pores that improve the performance of the materials as electrodes of VRFB are larger (macropores), since an increase in macropore volume reduces the resistance to flow and facilitate the access of the active species to the electrode surface.

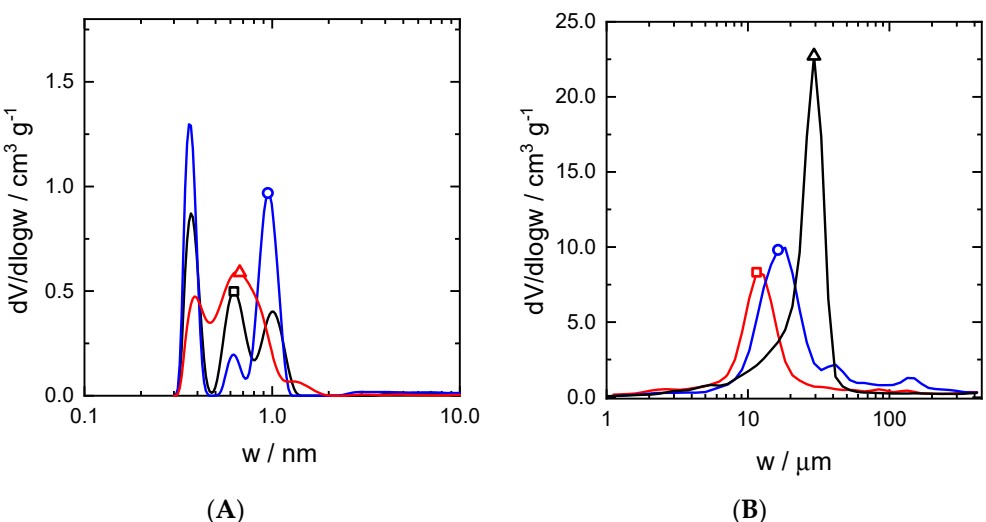

(A)                    (B)

**Figure 3.** Pore size distributions, obtained: (**A**) By applying NLDFT to nitrogen isotherms, and (**B**) by mercury intrusion porosimetry, for samples SucNaCl0.80 (□), SucNaCl1.00 (○) and SucNaCl1.25 (△).

Apart from the micropores, the sucrose-based carbon monoliths contain two types of macropores which were evidenced by mercury porosimetry (Figure 3B) and by SEM micrographs (Figure 4). These macropores were formed by two different mechanisms and hence present significant differences of size. The smallest ones, roughly spherical with diameters typically ranging from 10 to 30 μm, appeared when the NaCl crystals were removed from the structure during the washing step (Step 6 in Figure 1), i.e., between the first and the second pyrolysis. These pores are clearly visible in Figure 4, and the

detailed microphotograph for sample SucNaCl1.25 (Figure 4D) shows that they are very regular in size within each sample. Both the size and the total volume of these macropores increased with the proportion of NaCl in the initial mixture, i.e., from SucNaCl1.25 to SucNaCl0.80, leading to the corresponding decrease of bulk density (Table 1). The other types of macropores are large (several hundreds of microns) and very uneven in size and shape; as observed in Figure 4A–C they are not spherical pores, but irregular cracks, caused by the evolution of steam during the polycondensation of sucrose in the hot press. As the latter process is less controlled than the former one, these pores are much more heterogeneous in size, and there is no clear relationship between their average width and the initial formulation.

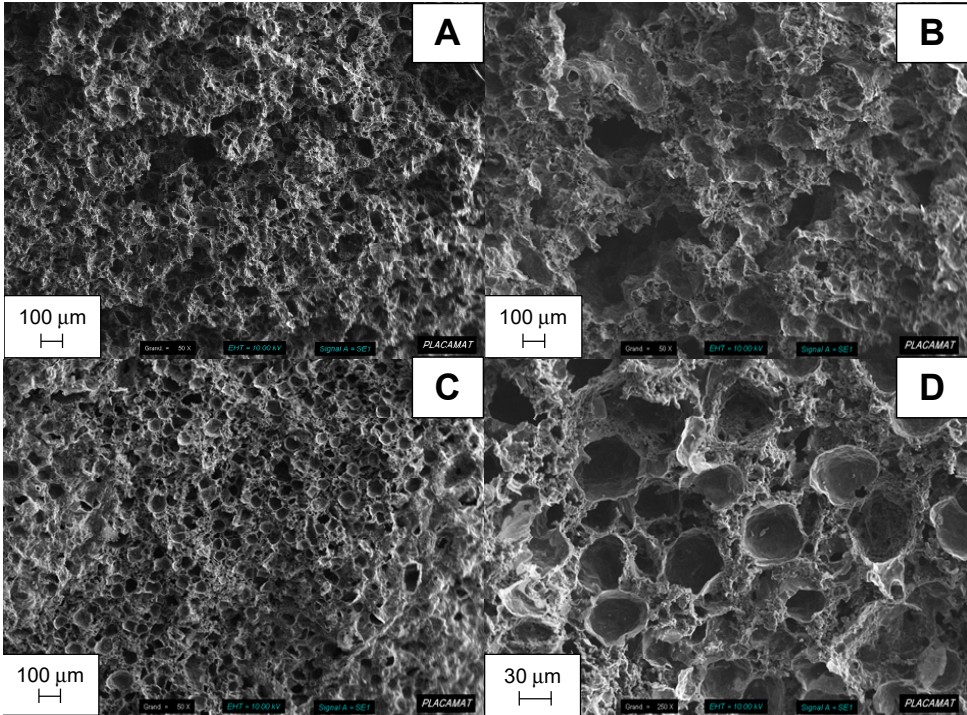

**Figure 4.** Scanning Electron Microscopy images of the samples: (**A**) SucNaCl0.80; (**B**) SucNaCl1.00; (**C**,**D**) SucNaCl1.25.

### 3.2. Mechanical Properties

The stress–strain curves of the three kinds of materials submitted to compression are presented in Figure 5A, and the corresponding changes of modulus and strength are shown in Figure 5B as a function of bulk density. The differences of maximum compression strength along the series of samples are in the range of the experimental error, so no clear conclusions can be obtained from them (being around 1.2 MPa for all of them). Regarding Young's modulus, this parameter slightly increased with the bulk density (from 94 MPa for SucNaCl0.8 to 117 MPa for SucNaCl1.25), as expected, since the materials become stiffer when their porosity decreases. The values reported in Figure 5B show that these hierarchical porous carbons are extremely strong with respect to their bulk density, and their mechanical properties are sufficient for the considered application.

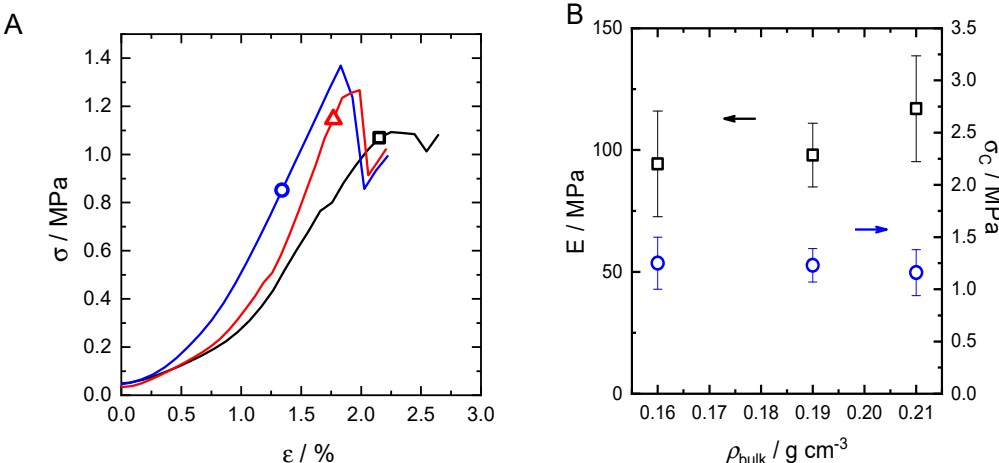

**Figure 5.** (**A**) Strain–stress curves of the materials submitted to compression: SucNaCl0.80 (□), SucNaCl1.00 (○), and SucNaCl1.25 (△). (**B**) Corresponding compression strength (○) and Young's modulus (□) as a function of porosity.

### 3.3. Carbon Nanotexture and Composition

Raman spectra are given in Figure 6. While the samples presented very different porosities, all spectra were nearly identical in terms of peak position, full width at half maximum and relative intensities. This suggests that the porosity has no effect on the proportions of disordered and graphite-like carbon in the materials. Indeed, the two main bands that can be observed are centred on 1350 cm$^{-1}$ and 1600 cm$^{-1}$, and are known as D band and G band, respectively [61,62]. The former is related to the presence of defects in the graphitic structure, while the latter is characteristic of sp$^2$ graphitic carbon. Therefore, a common way to evaluate the structural order in carbon materials consists in calculating the D to G intensity ratio ($I_D/I_G$). This parameter, which is related to the degree of graphitisation of carbon materials, did not change significantly along the series and it was slightly higher than 0.85 for all tested samples (Table 2). Furthermore, all the spectra showed a third band centred on 2700 cm$^{-1}$, which originates from the same vibration as for the D band and appears at twice its wavenumber, so it is commonly denoted as 2D band. The intensity of this band was also very similar in all spectra. The widths of both the D and G bands, the shallow valley between them and the poorly structured 2D band definitely suggest that these carbons are far from being graphitic, as expected for carbons derived from sucrose and heat-treated at temperatures not higher than 900 °C.

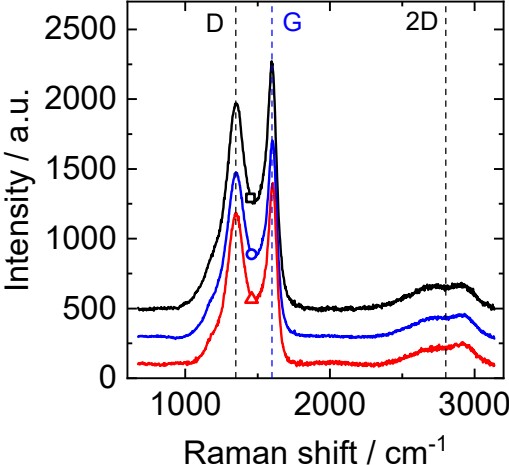

**Figure 6.** Raman spectra of the samples SucNaCl0.80 (□), SucNaCl1.00 (○) and SucNaCl1.25 (△).

**Table 2.** Carbon nanotexture (from Raman spectroscopy), and weight % of C, H and O (from elemental analysis).

| Sample | D Band cm$^{-1}$ | G Band cm$^{-1}$ | $I_D/I_G$ | %C wt.% | %H wt.% | %O wt.% |
|---|---|---|---|---|---|---|
| SucNaCl0.80 | 1353 | 1597 | 0.87 | 95.2 | 0.4 | 2.1 |
| SucNaCl1.00 | 1353 | 1602 | 0.87 | 96.0 | 0.4 | 1.8 |
| SucNaCl1.25 | 1347 | 1605 | 0.85 | 97.4 | 0.5 | 1.5 |

Elemental analysis of the carbon monoliths revealed no significant differences between the samples tested (Table 2). With such technique, differences less than 0.5 wt.% for C and O are indeed within the experimental uncertainty and can thus be considered as negligible. Moreover, the H/C ratio ranged from 0.05 to 0.06, suggesting negligibly small changes of aromaticity of the carbon texture. Such results thus confirm the aforementioned conclusion derived from Raman spectroscopy, according to which the texture and the chemical composition of the materials are independent from the sucrose to NaCl ratio in the precursor mixture. Therefore, any change of electrochemical behaviour should be explained based on differences of total porosity and porous structure.

### 3.4. Electrochemical Properties

Cyclic voltammetry was performed to evaluate the ability of the materials to promote the redox reactions of interest for a VRFB. Two important parameters are used to describe it: (i) The intensity of the current peaks relative to the reactions involving V(IV)/V(V) and V(II)/V(III) on the one hand; and (ii) the differences between peak potentials for each redox couple on the other hand. The latter parameter is indicative of the reversibility of the process, which is of major importance for understanding changes of battery efficiency.

Voltammograms depicted in Figure 7A revealed marked differences. First of all, it should be noticed that the four peaks corresponding to vanadium electrochemistry (V(IV) and V(II) oxidation, and V(V) and V(III) reduction) can be observed. Nevertheless, samples prepared with increasing sucrose fractions presented a significant capacitive response, making the voltammograms more and more square-shaped at potentials where there are no signals from redox reactions. This feature is related to the microporosity of the electrodes, which indeed increases from SucNaCl0.80 to SucNaCl1.25. Accordingly, the peaks corresponding to vanadium redox reactions are more easily observed for the least microporous sample, SucNaCl0.80, while they are not so obvious in the CV curves of the other two samples. Moreover, the potential differences between reduction and oxidation reactions for the two redox couples (V(V)/V(IV) and V(III)/V(II)) is minimum for SucNaCl0.80 and it increases for SucNaCl1.00 and SucNaCl1.25.

This variation is assigned to mass transfer resistance of ions from the electrolyte to the active sites at the electrode surface, since a clear correlation between the value of these differences and the porosity of the samples measured by mercury intrusion porosimetry was found (Figure 7B). Voltage efficiency of the battery is closely related to these potential differences since the reactions involved are those occurring when charging and discharging the VRFB. In other words, the lower the difference between redox potentials, the more similar are the electrochemical potential, and thus the energies involved during charge and discharge.

Once the materials were thus proved to promote charge transfer reactions for reducing and oxidising vanadium species present in the electrolyte, they were tested in an RFB. Prior to the charge–discharge tests, a polarization curve was obtained for each material (Figure 8A). At given potential values, the highest current densities were found for the sample SucNaCl1.25, followed by SucNaCl1.00 and SucNaCl0.80. This order is related to the different capacitive behaviours discussed above.

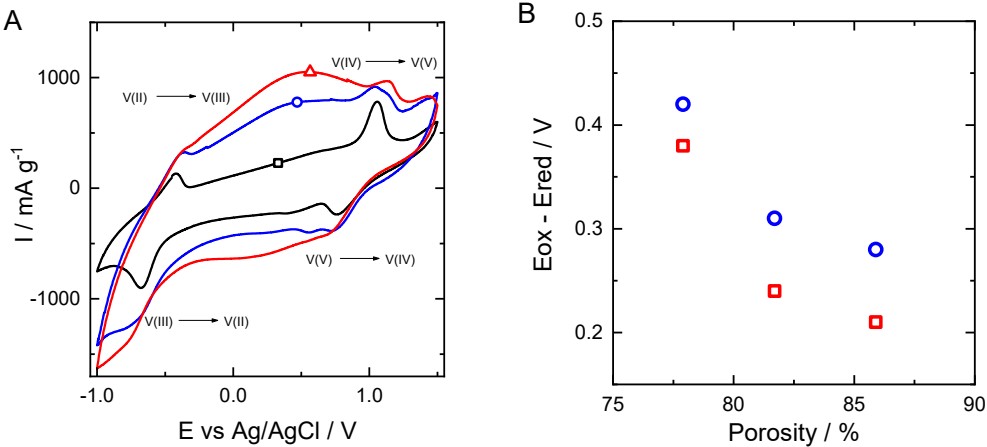

**Figure 7.** (**A**) Cyclic voltammograms recorded at 5 mV·s$^{-1}$ in 0.15 M V(III)/V(IV) solution prepared in 3 M H$_2$SO$_4$ for samples SucNaCl0.80 (□), SucNaCl1.00 (○) and SucNaCl1.25 (△). (**B**) Differences between oxidation and reduction potential as a function of porosity: V(V)/V(IV) (○) and V(III)/V(II) (□).

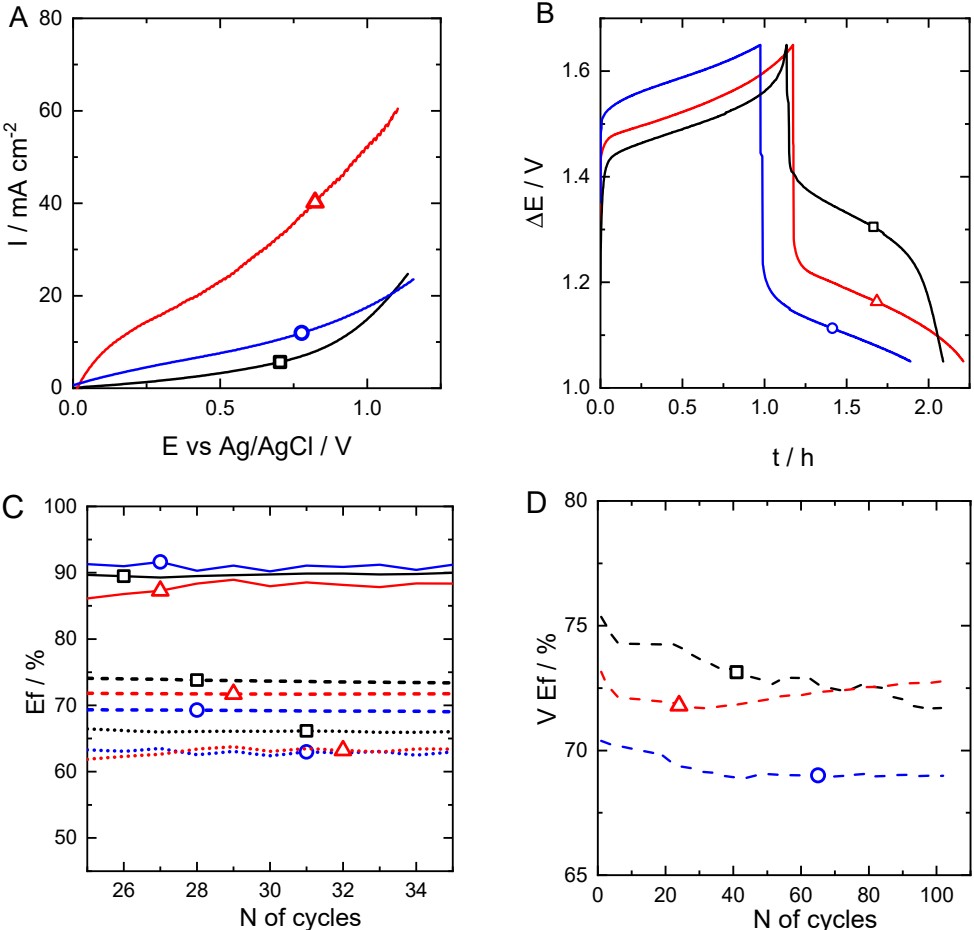

**Figure 8.** (**A**) Polarisation curves. (**B**) Charge–discharge curves of the same materials for one cycle at 20 mA·cm$^{-2}$ in 20 mL/half-cell of a 1.5 M V(III)/V(IV) solution prepared in 3 M H$_2$SO$_4$. (**C**) Coulombic (solid line), energy (dotted line) and voltage (dashed line) efficiencies along the first ten cycles at 20 mA·cm$^{-2}$ after cycling at 50 and 80 mA·cm$^{-2}$, and (**D**) voltage efficiencies for one hundred cycles for samples SucNaCl0.80 (□), SucNaCl1.00 (○) and SucNaCl1.25 (△).

Charge–discharge experiments were performed at several current densities and the results for one cycle at 20 mA·cm$^{-2}$ showed very similar results in terms of charge and discharge time, which is related to the battery capacity since the current density was constant and the same for all the samples (Figure 8B). This means there is not a great difference in terms of electrolyte utilization along the tested samples. Nevertheless, differences regarding charge and discharge voltage of the curves are clearly observed, given that the polarization is much lower in the case of SucNaCl0.80 than for the other two samples, which is in good agreement with cyclic voltammetry experiments. Even though potential limits were increased for the cycles obtained at 50 and 80 mA·cm$^{-2}$, charge and discharge occurred very fast. Therefore, efficiencies could be calculated only at a current density of 20 mA·cm$^{-2}$ (Figure 8C, Table 3).

**Table 3.** Coulombic (Q), energy (E) and voltage (V) efficiencies (Ef) for the charge–discharge of the VRFB at a current density of 20 mA·cm$^{-2}$ (calculated as the average of the ten first cycles after GCD at several current densities).

| Sample | Q Ef % | V Ef % | E Ef % |
|---|---|---|---|
| SucNaCl0.80 | 91 | 74 | 67 |
| SucNaCl1.00 | 92 | 69 | 63 |
| SucNaCl1.25 | 90 | 71 | 64 |
| Activated CF [63] | 92 | 85 | 78 |

In this sense, reported coulombic efficiencies are very similar for the three materials tested and are around 90%, which is not surprising, since this parameter is more related to other characteristics of the system such as membrane, inter-electrode distance and flow rate. Differences are, however, more pronounced when variations of energy and voltage efficiencies are considered. Those differences are related to the reversibility of the process already described above, i.e., they increase with the porosity of the electrode (Table 3). As it can be observed, these values are still far from the ones obtained by our group for commercial carbon felt (tested in the same experimental set-up employed in this work [58]), but they are promising, considering that this is the first time this kind of material has been used in VRFB. Furthermore, the results can be improved by optimizing the synthesis procedure and through functionalization of material surface as we did in the above-mentioned work [63]. Values of voltage efficiencies are globally maintained along the experiments for samples SucNaCl1.00 and SucNaCl1.25, whereas they decrease with the number of cycles in the case of SucNaCl0.80, from 75 to 72% (Figure 8D), probably due to the more extensive degradation of this material compared to the others, which was enhanced by its larger macropore volume and lower mechanical resistance. Therefore, a compromise between reactivity and long-term stability is desirable to choose the best material as electrode for VRFBs.

## 4. Conclusions

Carbon materials with hierarchical porosity were obtained from sucrose using sodium chloride as hard template. Samples were synthesised with various sucrose to NaCl weight ratios in the initial mixture to get materials with different porosities. Porosity, carbon texture and chemical composition were evaluated by different techniques, and the materials were then tested as electrodes for VRFBs.

While the surface chemistry of the materials was very similar, variations of pore volume and pore size distributions were evidenced as the sucrose to NaCl proportion increased. In this way, sucrose-rich formulations led to higher amounts of carbon after pyrolysis, and hence to samples presenting higher micropore volumes; in contrast, salt-rich formulations led to larger macropore volumes once the salt, acting as a porogen, was leached out by extensive washing.

Variations of porosity were correlated with the electrochemical behaviour of the materials, and a good agreement between capacitive response and microporosity was observed. Regarding vanadium electrochemistry, differences in oxidation and reduction reaction potentials for both V(V)/V(IV) and V(III)/V(II) decreased as the macropore volume increased, so that the reversibility of the redox processes involved in the VRFB charge/discharge was improved. Reversibility is also related to the battery performances characterised by their efficiency values. While coulombic efficiencies were very similar, energy and voltage efficiencies in the first cycles were higher for the sample with the largest macropore volume (SucNaCl0.80), which is also the one with the lowest differences between oxidation and reduction potentials for both redox couples. Nevertheless, after several charge/discharge cycles, the voltage efficiencies of the other two samples were maintained, while a decrease was observed for SucNaCl0.80, revealing the degradation of this material with time. Furthermore, the higher mechanical stability of SucNaCl1.00 and SucNaCl1.25 should allow changing the design of the electrode in such a way that efficiencies can be further improved. Finding a compromise between mechanical and chemical stability of the material and its electrochemical performances will be considered in a forthcoming work. This work can be seen as the very first time that very abundant bioresource raw materials (such as sucrose and NaCl) have been used for fabricating VRFB electrodes, and this fact represents a new range of possibilities to explore more sustainable materials at a lower cost than the ones commonly used for this application.

**Author Contributions:** Conceptualization, M.E., N.B. and A.D.; methodology, J.F.V.-V., B.K.; validation, M.E. and R.E.H.; formal analysis, J.F.V.-V., B.K., A.C., V.F. and M.E.; investigation, J.F.V.-V. and B.K.; resources, A.C., V.F. and M.E.; data curation, J.F.V.-V., B.K., V.F. and M.E.; writing—original draft preparation, J.F.V.-V.; writing—review and editing, R.E.H., V.F., A.C. and M.E.; visualization, M.E.; supervision, A.C., N.B., A.D., M.E.; project administration, M.E.; funding acquisition, A.C., V.F. and M.E. All authors have read and agreed to the published version of the manuscript.

**Funding:** This work was supported by ICEEL and Region Grand Est and J.F.V.-V. was hired with these fundings. This work was partly supported by a grant overseen by the French National Research Agency (Pc2TES ANR-16-CE06-0012-01), and the authors involved in it (AC, BK and VF) acknowledge the support of the project's coordinator, Mrs Fouzia Achchaq. This study was partly supported by TALiSMAN project (2019-000214), funded by European Regional Development Fund (ERDF).

**Institutional Review Board Statement:** Not applicable.

**Informed Consent Statement:** Not applicable.

**Data Availability Statement:** Not applicable.

**Acknowledgments:** The authors acknowledge ICEEL and Region Grand Est, French National Research Agency and European Regional Development Fund for their financial support. They are grateful to Manuel Dossot for the Raman spectroscopy measurements, to Philippe Gadonneix for the elemental analysis and his kind help with other techniques, and to Philippe Legros from PLACAMAT—Bordeaux for SEM microphotographs.

**Conflicts of Interest:** The authors declare no conflict of interest.

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
