# Peer review of "Carbon Monoliths with Hierarchical Porous Structure for All-Vanadium Redox Flow Batteries"

_batteries, doi:10.3390/batteries7030055_

Round 1
Reviewer 1 Report
The reviewer would like to thank the authors for the submission of the manuscript titled "Carbon Monoliths with Hierarchical Porous Structure for All-Vanadium Redox Flow Batteries".
While various carbon structures, derived from both natural sources and artificially prepared have been investigated for energy storage and carbon capture, the authors provide a clear motivation for the current work in the introduction.
The methods, results and discussion provide clear explanations that are supported by the experiments. Overall, the manuscript is very well written and will benefit the readers.
A few minor comments are added here:
- Figure 8c, the voltage efficiencies (dashed line) are not clearly visibly. Could the authors modify the symbol size or line for easy visibility.
- Figure 8c, could the authors provide an explanation for the cycle# starting at 25 as the first of the 10 cycles, as mentioned in the caption.
- Figure 8 c, could the authors provide an explanation for the increase in the coulombic and voltage efficiency and voltage for SuNaCl1.25 from cycle 25-29 and in Figure 8d from cycle 40-100.
- A comparison of the performance with other carbon based electrodes for redox flow batteries would benefit the reader. Could the authors provide another plot comparing the efficiency of their work with other similar work in literature.
Reviewer 2 Report
The manuscript reported the preparation of carbon electrodes with hierarchical porosity from sucrose and based on sodium chloride as the hard template. Various physical-chemical parameters, including porosity, texture, chemical composition were characterized and evaluated. They were also tested as electrodes for VRFBs.
Variations of porosity were correlated with the electrochemical behavior of the materials. As the macropore volume increased, the reversibility of the redox processes involved in the VRFB charge/discharge was improved. Nevertheless, after several charge-discharge cycles, the degradation of certain electrodes takes place with time. In the future, finding a compromise between the mechanical and chemical stability of the electrode and its electrochemical performances will be vital for the improvement.
I consider the content of this manuscript will definitely meet the reading interests of the readers of Batteries journal. More importantly, this article only uses sucrose and salt, which are the most widely existing in nature, as raw materials to prepare VRFB electrodes, which can be said to open a door to the future of low-cost and easy-to-prepare carbon electrodes. This can really be regarded as a gift from nature, and it is a very interesting result.
However, I have to point out, the characterization should be better compared with one commercial electrode, at least in the electrochemical performance part. Indeed, if in mechanical and porosity property part, also some comparisons are made will be perfect. Although the performance in VRFB has not reached a perfect level due to unknown reasons leads to the relatively low VE, the overall characterization and analysis of this manuscript are still comprehensive and clear.
This article can be used as a kind of basic research and preliminary exploration of new synthesis methods, to give readers the enlightenment of innovative thinking. We should not be entangled in this article, the electrode must have extremely high battery performance. I believe that future improvements based on this approach/electrode will help to achieve a higher performance of VRFB.
Therefore, I suggest giving a minor revision and the authors need to clarify some issues or supply some more data to enrich the content. My detailed comments can be found in a separate PDF file.

Reviewer 3 Report
In this paper, monolithic carbon materials based on sucrose polycondensed in the presence of NaCl were prepared with a protocol like the one described elsewhere. In the preparation process, both sucrose and sodium Chlo-riders were ball-milled together and molten into a paste that was hot-pressed to achieve Polycon-sensation of sucrose into a hard monolith. After that, their relevance to promote vanadium redox reactions was evaluated by cyclic Voltammetry. The idea is interesting, but there are still some problems that will be improved after modification:
(1) The description of Figure 4 needs to be amended.
(2) A few grammar and scientific notation problems in this article need to be modified, and it will be better to read them after modification.
(3) The experimental steps in this article need to be more detailed.
